# Prediction of Bending Properties for 3D-Printed Carbon Fibre/Epoxy Composites with Several Processing Parameters Using ANN and Statistical Methods

**DOI:** 10.3390/polym14173668

**Published:** 2022-09-04

**Authors:** Francisco M. Monticeli, Roberta M. Neves, Heitor L. Ornaghi, José Humberto S. Almeida

**Affiliations:** 1Department of Aeronautical Engineering, Technological Institute of Aeronautics (ITA), São José dos Campos 30161-970, Brazil; 2PGPROTEC, University of Caxias do Sul, Caxias do Sul 95070-560, Brazil; 3Mantova Indústria de Plásticos Ltda, Caxias do Sul 95045-137, Brazil; 4School of Mechanical and Aerospace Engineering, Queen’s University Belfast, Belfast BT9 5AH J, UK

**Keywords:** 3D printing, CFRP, artificial neural network, statistical approach

## Abstract

The effects of processing parameters on conventional molding techniques are well-known. However, the fabrication of a carbon fibre (CF)/epoxy composite via additive manufacturing (AM) is in the early development stages relative to fabrications based on resin infusion. Accordingly, we introduce predictions of the flexural strength, modulus, and strain for high-performance 3D printable CF/epoxy composites. The data prediction is analyzed using approaches based on an artificial neural network, analysis of variance, and a response surface methodology. The predicted results present high reliability and low error level, getting closer to experimental results. Different input data can be included in the system with the trained neural network, allowing for the prediction of different output parameters. The following factors that influence the AM composite processing were considered: vacuum pressure, printing speed, curing temperature, printing space, and thickness. We further demonstrate fast and streamlined fabrications of various composite materials with tailor-made properties, as the influence of each processing parameter on the desirable properties.

## 1. Introduction

Carbon fibre-reinforced polymers (CFRPs) are an attractive alternative for the aeronautical industry, as lightweight composite materials can replace conventional metallic components [1,2]. With the industrial growth of unmanned aerial vehicles (UAVs) and the emergence of urban air mobility, the application of CFRP structures has substantially increased, especially when complex geometries and shapes [3,4,5] are required and weight minimisation is vital. The progress in additive manufacturing (AM) is an interesting option regarding weight reduction in CFRPs, e.g., providing complex 3D geometries without sacrificing structural performance [1,6,7].

Ming et al. [6] exhibited complex structures (pentagram and honeycomb structures) manufactured from carbon fibre-reinforced thermoset composites through AM. They allow the application of high specific stiffness in structural components that could not be achieved with conventional manufacturing approaches. Azarov et al. [7] demonstrated the application of a continuous fibre composite processed by 3D printing to manufacture a small UAV frame and ensured adequate mechanical properties in the complex geometry with a 60% weight reduction. In addition, Sano et al. [8] exhibited a better quality complex component by using a low fibre fraction while improving the processing aspects to increase the fibre volume fraction of the printed composite in a thermoset matrix. Shi et al. [9] proposed a capillary-driven model to produce AM composites based on viscosity control and degree of curing, increasing tensile strength close to conventionally manufactured composites.

Gnanasekaran et al. [10] demonstrated that the conventional AM method, namely fused deposition modelling (FDM), can manufacture low-cost and functional composite structures. The 3D printing with abrasives, such as carbon nanotubes and graphene, demonstrated improvements in elastic behaviour and conductive properties of thermoplastic matrices, with the risk of greater nozzle wear [10]. This is also confirmed for thermosetting matrix in the Jiang et al. [11] review, in which high content of graphene limits the AM, conducting in discontinuous extrusion process (regarding polymer agglutination with reinforcement).

The AM process is as important as the reinforcement and matrix used for the conventional polymer composite manufacturing method [12,13,14]. Fused filament fabrication (FFF) is a method that presents the possibility of employing continuous fibres pre-impregnated with a thermosetting matrix, depositing thin lines adjacent to each other [15,16]. Meanwhile, imperfections such as porosity are detrimental to the mechanical behaviour of the final component [17,18]. The region between the printed filaments must ensure proper interfacial adhesion, as an FFF usually presents a low shear and delamination resistance, owing to the free spacing (voids) between the filaments [17,18,19,20]. The FFF processing control is directly associated with the mechanical properties generated by CFRPs and is even more evident for structures with complex geometries, as using the appropriate processing parameters can reduce the formation of defects [21,22].

The processing parameters are responsible for preventing defects and ensuring good adhesion between the matrix and reinforcement and between the printing filaments, thereby reducing void formation [23,24,25]. In addition, the printed composite quality directly contributes to the mechanical properties of the material, as inappropriate interfacial adhesions and high void contents are detrimental to mechanical performance [25,26,27].

The finite element method (FEM) could be used to analyse the processing and mechanical behaviour of the interface between printed sections [28,29,30,31]. For instance, Zhang et al. [32] proposed a new concept for continuous 3D printed curved fibres. They applied the main stress trajectories using FEM for open-hole specimens. Alternatively, Moradi et al. [33] exhibited the influence of printing parameters on mechanical behaviour based on the design of experiments (DOE) method. They used carbon fibre-reinforced polylactic acid (PLA) processed by fused deposition modelling (FDM), setting the optimal parameters to maximize failure load. On the other hand, Meiabadi et al. [34] studied the combination of DOE and artificial neural network (ANN) to estimate the toughness (mechanical performance) of AM PLA composite. Deep learning (DL) is a broader family of machine learning (ML) methods based on learning representations of data [35,36]. The aim is to make better representations and create models based on interpreting information processing and communication patterns based on nervous systems, such as ANN [37,38,39]. Artificial network analysis has been demonstrated to be a feasible estimation of process–structure–behaviour properties, saving project cost and time [34,35,40]. Several researchers have applied the statistical methodology to provide the prediction behaviour of AM composites [40,41,42]. 

The majority of works in the literature use thermoplastic matrices since thermoset ones add high complexity to controlling the printing process. The same methods and processing parameters used for thermoplastic matrices could not be directly applied to thermoset ones, once the curing kinetics are entirely different, requiring different levels of processing control. To the best of the authors’ knowledge, there is a gap in the literature on process/property optimization of AM thermoset composites, as shown in the review performed in Ref. [18]. Processing control is critical for thermoset matrices since the parameters applied during 3D printing contribute to the printed filament sizing and the interlayer bonding [6,43,44]. Therefore, this work reports the main influence of the FFF 3D-printing parameters on the mechanical behaviour of the parts, considering a continuous carbon fibre thermosetting polymer composite.

This work aims to predict the process–structure–behaviour properties for CFRPs processed by AM, combining ANN and DOE methods. For that purpose, the predictive approach was carried out based on several experimental flexural mechanical tests as a function of parameter level variations. The contribution to the field is the characterization and prediction of the flexural mechanical behaviours (strength, modulus, and strain) of a continuous CFRP thermoset composite FFF based on printing parameter variations.

## 2. Additive Manufacturing Processing Parameters

Additive manufacturing is a promising method to generate components in complex geometries and expand the application of polymer composites with structural requirements [43,45]. Thermoset polymers exhibit superior thermal/mechanical stability compared to most other matrices, producing suitable material for structural applications [44]. Nevertheless, thermoset polymer presents are complex to control, such as viscosity, cure temperature/time, pressure, and printing stability, among others.

Figure 1 illustrates the FFF scheme and the direction of the CFRP printing composite. The printing speed is responsible for the velocity of the printing nozzle in the two-dimensional plane. The printing space is the center distance between the adjacent printed fibres (Figure 1a) and can create a printing overlay (Figure 1d) to avoid porosity between the printing filaments (Figure 1e). The distance can be determined by the nozzle diameter and printing speed, which can increase or decrease the printing filament diameter. The printing speed also determines the impregnated fibre feeding speed; this affects the imprinting flow and, consequently, the printing diameter (Figure 1b,c). The thickness of the space can also contribute to printing overlay control, as it determines the space between the adjacent filaments in the z-direction.

The aforementioned processing parameters change the printed diameter, and each level combination controls the printing overlay. The printing overlay can strengthen the bonding to enhance the shear between the printed filaments, ensuring high mechanical performance [6]. However, an inappropriate overlay can induce defects, produce fibre misalignments, and change the material sizes and mechanical behaviours [18].

The temperature affects the viscosity, polymerization rate, and crosslinking reactions (i.e., kinetic energy), similar to the case with conventional thermoset composite manufacturing [46,47,48]. In addition, the vacuum pressure can remove the volatiles and moisture generated during the curing process. The appropriate cure temperature and vacuum pressure levels can also reduce the porosity and internal residual stress of the material generated by the crosslinking of the thermoset matrix, which affects the mechanical performance [46,47].

## 3. Materials and Methods

### 3.1. Experimental Details

Continuous carbon fibre from Tenax^®^-J HTS40 (Toho Tenax, Tokyo, Japan) was used as the reinforcement, and the epoxy resin DER 671 and a curing agent DICY (95/5 wt%) were used as the matrix system, from Dow, Pittsburg, CA, USA. The composite filament was manufactured before printing and deposited in rollers, which were used to print the composite. The process was conducted using FFF-based 3D printing (Xi’an, China), in which the reinforcement was impregnated with the epoxy system to manufacture the composite part, as shown in Figure 2. The filament was printed unidirectionally. Additional details regarding the fibre characteristics can be found in [19].

### 3.2. Statistical Approach

Regarding the large range of the parameter level combinations, a design of experiments approach was conducted to reduce the number of experiments. The orthogonal Taguchi [49] array L_25_ was applied, as shown in Table 1, considering the printing speed, space, thickness, cure temperature, and vacuum pressure parameter levels. The levels were chosen as the maximal, minimal, and intermediates levels.

Flexural tests (three-point bending) were conducted on five specimens for each experimental number (Table 1). Flexural behaviour is an important parameter for industrial application requirements, which relates to material stiffness and shear behaviour in the inter-printing section. The flexural strength, modulus, and strain were measured using the universal testing machine from MTS Systems, following the American Society for Testing and Materials D7264/D7264M–07 standard. The specimen dimensions were 250 × 25 × 2 mm^3^. The mechanical test was conducted with a displacement rate of 2.0 mm·min^−1^. Additional details regarding the mechanical test procedure are provided in the Reference [6].

An analysis of variance (ANOVA) was performed to measure the contribution of each parameter and to determine the null hypothesis validation. The H0 hypothesis considered no significant difference between parameter level variations, and the H1 hypothesis was that the parameter level variation influenced mechanical performance. The ANOVA procedure was conducted using Minitab18 for a single factor, with α = 0.05 and 95% reliability.

### 3.3. Artificial Neural Network (ANN) Methodology

The ANN is inspired by the nervous system, in which neurons represent processing elements for performing operations parallel to data processing [50]. Literature works [51,52] have shown that the ANN approach is used to predict the processing design of polymer composite, saving time, effort, and project cost. ANN models are widely used in medical/biological sciences and data systems, presenting the opportunity to model process–structure–behaviour for engineering applications [53]. The proposed method may generate a prediction modelling of the composite mechanical behaviour processed by AM and provide a database for different input parameters. Thus, reducing resource consumption and enabling the expansion of the proposed composite in components with real structural applications.

Figure 3 shows an ANN scheme consisting of input, output, and hidden layers. In this study, the inputs are the processing parameters, and the outputs are the mechanical properties. Figure 4 shows the flowchart of the ANN training process following the structure established in this work.

To calculate each prediction data point, an activation function needs to be applied to the input data (as related to the complex relationship between the input variables and output terms). The hyperbolic tangent (Equation (1)) has been used in multi-layer neural networks as the activation function for complex experimental data combinations [54]. Here, the weighting algorithm was resilient backpropagation with backtracking, in which the error function was the sum of the squared errors, and the threshold of the error function was 0.01. The error was characterized by the *R*^2^ coefficient (Equation (2)) [52,55].
(1)$$f\left(x\right)=\frac{e^{x}-e^{-x}}{e^{x}+e^{-x}}$$
(2)$$R^{2}=1-\frac{\sum_{i=1}^{n}{\left(x_{pr}{}^{\left(i\right)}-x^{\left(i\right)}\right)}^{2}}{\sum_{i=1}^{n}{\left(x^{\left(i\right)}-\bar{x}\right)}^{2}}$$
where $x_{pr}{}^{\left(i\right)}$ is the predicted data, $x^{\left(i\right)}$ represents the experimental data, $\bar{x}$ is the mean value of $x^{\left(i\right)}$, and n is the number of experimental data (i.e., 25).

The neural network algorithm used was resilient backpropagation with backtracking to solve the optimization process, following output values for all neurons in hidden layers according to Equation (3), which makes a fast adjustment to the weight function according to the error found in each iteration. The limitations are the dependency on the input data and sensitivity to noisy data. However, the presented material showed good reproducibility with a low standard deviation, in which all the main factors were considered input data.

During the training, 100% of the data points in Table 1 were selected by the K-fold function for training, and extra experimental data were performed for validation. The training repetition was performed until the mean square errors (MSE) of 0.0001—Equation (4)—were minimised. The number of hidden layers was 5 and 10 neurons in each layer.
(3)$$f\left(x\right)=y_{j}=f\left(\sum_{i}w_{ij}\cdot y_{i}\right)$$
(4)$$MSE=\frac{1}{n}\overset{T}{\underset{i=1}{\sum }}{\left(x_{pr}{}^{\left(i\right)}-x^{\left(i\right)}\;\right)}^{2}$$
where *y_i_* is the output, *i_th_* is the neuron in the layer considered, *y_j_* is the output in the *j_th_* neuron layer, *f(x)* is the sigmoidal function, *T* is the number of predictions, and *t* is interaction.

### 3.4. Response Surface Methodology (RSM)

The RSM approach was carried out to predict mechanical behaviour using different levels for each parameter not accessed experimentally. Equation (5) describes the interactions between the processing parameter levels (printing speed—*P_v_*, space—*P_s_*, thickness—*t*, cure temperature—*T*, and vacuum pressure—*P*) and mechanical properties (flexural strength—*σ*, modulus—*E*, and strain—*ε*). The predicted data from the ANN and experimental results were used using Equation (5) to refine statistical regression, thereby providing a three-dimensional prediction approach.
(5)$$Z=\beta_{0}+\overset{k}{\underset{i=1}{\sum }}\beta_{i}x_{i}+\overset{k}{\underset{i=1}{\sum }}\beta_{ii}x_{i}^{2}+\overset{k}{\underset{j=1}{\sum }}\beta_{j}y_{j}+\overset{k}{\underset{j=1}{\sum }}\beta_{jj}y_{j}^{2}+\;\overset{k}{\underset{j=i}{\sum }}\beta_{ji}x_{i}y_{j}$$

Here, Z represents the predicted response, that is, the mechanical behaviour (*σ*—MPa, *E*—GPa, and *s*—%). *x_i_* and *y_j_* are the parameter levels associated with printing speed *Pv* (in mm·min^−1^); space *Ps* (in mm); thickness *t* (in mm); cure temperature *T* (in °C); and vacuum pressure *P* (in MPa). The parameter $\beta_{0}$ is a constant coefficient, $\beta_{i}$ and $\beta_{j}$ are the linear coefficients, and $\beta_{ij}$ is an interaction coefficient.

## 4. Results and Discussion

### 4.1. Analysis of Variance (ANOVA) Results

The experimental data were treated via the ANOVA (data presented in Table 2) to quantify the influence of each parameter on the mechanical behaviours. According to Montgomery [49], the null hypothesis (*H*_0_) is true if *F* > *F_critical_* and *p*-value < 0.05. All parameters evaluated present a false null hypothesis, validating hypothesis *H*_1_, as *F > F_critical_* indicates that the oscillation of each parameter level directly influences the mechanical behaviour of the material. This statement is confirmed by the *p*-value < 0.05, which guarantees that each parameter influences the responses with high reliability.

According to the *F* value distribution, Figure 5 represents the percentage contribution of each parameter to the mechanical behaviour of a CFRP processed by AM. For all mechanical properties, the vacuum pressure parameters show a high contribution factor, as it ensures the removal of the residual tension, volatiles, and other defects formed during polymer matrix curing. Both factors also strongly influence conventional composites [56,57,58]. The curing temperature affects the curing speed of the thermoset matrix, which can generate residual stress and incomplete bonds at an inappropriate temperature [59,60,61]. Thus, the stresses generated during the curing stage will directly influence the mechanical behaviour of the material, mainly at intra-imprint interfacial strength [56].

The printing speed has a low percentage contribution, as the control of the printing distance between the filaments (in horizontal and thickness directions) reduces the influence of this parameter. This occurs due to the distance between the filaments covering porosity gaps and the printing overlay provided by stretching the filament at high speed or increasing the filament diameter at low speeds. Printing speed variation can generate the stretching or contraction of the printing section (changing printing diameter), as explained in Section 2. However, this variation can be supplied by the control of the horizontal and vertical spacing between the filaments, reducing the experimental influence of the printing speed.

The printing space and thickness are the most responsible for reducing the spaces between printed filaments (inter-filament porosity), thereby increasing the bonding between the printed filaments. As a result, both parameters (printing space and thickness) strongly influence the flexural strength and modulus, which results in the higher shear resistance between the imprinted filaments. Both parameters are mainly responsible for the laminate quality during processing, for example, the scratching, breakage, and warpage of the fibres. 

The strain properties are similar to flexural strength and modulus for the printing speed, thickness, and cure temperature. The variation of AM parameters results in differences in the mechanical properties of the material, as shown in this work. Nevertheless, the spacing generated between printed filaments results in a reduction of the maximum load capacity (strength), modifying the angle stress-strain curve (i.e., modulus), and maintaining the strain levels close to each other. Thus, the percentage of contribution (PC) for strain differs from the modulus response. On the other hand, strain is affected by curing parameters since they guarantee greater binding efficiency of macromolecules. Moreover, the flexural strain exhibits a lower influence on printing space than the printing thickness parameter, as the deformation in the thickness direction is more prone to the flexural test procedure. The printing spacing on the vertical axis (printing thickness) can result in a spacing between the layers along the thickness, generating a state of compressive deformation during the 3–point bending test, increasing the contribution of the printing thickness parameter, a behaviour that was similarly found by ref. [48].

The same effect occurs with vacuum pressure, in which the removal of volatiles, porosity, and other defects is responsible for controlling the deformation capacity of the material during the application of the bending load. Following these results (i.e., percentage of contribution—PC), it is possible to determine the most important parameter sequence to control, aiming to reduce defects.

### 4.2. ANN Prediction Data

As discussed above, the mechanical behaviours were predicted by applying a neural network. Table 3 presents the average of experimental results and prediction data associated with each parameter level, as listed in Table 1. The average error of the ANN method is 0.43%, ensuring a highly reliable prediction. The errors found were below the standard deviation between testing repetitions, indicating that the presented and questioned variations are more related to the intrinsic standard deviation of the proposed experiments than any specific combination that could provide greater or lower error during modelling. The ANN prediction data were obtained following processing parameter variations, in which the printing speed, space, thickness, cure temperature, and vacuum pressure were the input parameters, and the flexural strength, modulus, and strain were the outputs. The performance of ANN is shown in Appendix A.

The prediction procedure can be found in the Appendix A, in which different levels of process parameters can be applied to predict mechanical performances not otherwise accessed experimentally.

Figure 6 presents a neural network regression plot for each evaluated mechanical property. Figure 6a–c show the neural network regression, i.e., the interactions between the ANN prediction data and experimental processing datasets, in which the linear interaction exhibits insignificant variation. In addition, the linear function of the prediction output vs. experimental data confirms a good fit, with a high coefficient of determination *R^2^* > 0.99, as associated with the low errors found. The similar linearity between each mechanical property indicates a high degree of reliability, and the dispersion confirms the reproducibility of the ANN model for the mechanical behaviours as a function of the processing parameters.

Figure 6d–f exhibit the residual plots for the ANN prediction data. The probability vs. residual data shows a linear dispersion and regression trend, indicating that the error is normally distributed. The slope of the residual curve indicates the data dispersion; these are different for each curve because they represent mechanical properties with different orders of magnitude. The residual range is directly associated with the error, in which a lower range of residual data generates a decrease in the probability of the error occurrence. The ANN model investigation is suitable for mechanical behaviour forecasting for AM processing of CFRPs, as shown by the error values and linearity of the residual distribution.

### 4.3. Response Surface Methodology (SRM)

The FFF-based 3D printing of continuous CFRP composites has numerous parameter–level combinations. To minimise the experimental number of processed composites, a statistical approach was used to predict the mechanical behaviour of each parameter level combination [62,63]. For this purpose, the RSM was used to predict the flexural strength, modulus, and strain trends between the explanatory processing parameters. 

Figure 7 shows the RSM for the flexural strength (Figure 7a–c) and modulus (Figure 7d–f) based on the processing parameter levels, in which Equations (6)–(11) refer to Figure 7a–f, respectively. In addition, the black dots are represented by the ANN prediction values and experimental data. Considering the high number of factors (parameters), the possibilities of parameter combinations for each mechanical property are presented in the Appendix A. The coefficient of determination between the generated response surfaces and the experimental points showed a variation of *R*^2^ > 0.91.
(6)$$\sigma =274.9-0.45P_{v}+1187.2P_{s}+2.5\times 10^{-5}P_{v}^{2}-615.2P_{s}^{2}+0.33P_{v}P_{s}$$
(7)$$\sigma =247+715P_{s}+166t-209P_{s}^{2}+1307t^{2}-714P_{s}t$$
(8)$$\sigma =-5774.6+71.5T-4454.3P-0.2T^{2}-12484.7P^{2}+11.2TP$$
(9)$$E=96-0.037P_{v}-35.5P_{s}+3.9\times 10^{-6}P_{v}^{2}+6.9P_{s}^{2}+0.02P_{v}P_{s}$$
(10)$$E=13.2+32.6P_{s}+109t+6.6P_{s}^{2}+133.4t^{2}-141.6P_{s}t$$
(11)$$E=-533.3+6.5T+110.2P-0.02T^{2}+1582.1P^{2}-0.19TP$$
where *σ* (in MPa) represents the flexural strength, *E* (in GPa) the flexural modulus, *Pv* (in mm·min^−1^) printing speed, *Ps* (in mm) printing space, *t* (in mm) printing thickness, *T* (in °C) cure temperature and *P* (in MPa) is the vacuum pressure.

All of the parameters directly affect the mechanical behaviours of the composite, as mentioned previously. The combination of printing speed and space parameters (Figure 7a) exhibits interaction between the extensional flow, changing the diameter of the printed composite, evidencing the need to control the distance of the printed filament. 

The increase in printing speed will, inevitably, require a decrease in the distance between the filaments since the filament elongation due to high speed generates the need for printing filaments close to each other to prevent inter-voids formation. In addition, the printing thickness controls the distance between the stacking filaments in the z–direction, and the lower space creates a compression force through the composite thickness, reducing the printing quality as a function of the broken fibre and composite scratching.

The maximization of flexural strength leads to the lowest printing speed (i.e., 200–500 mm·min^−1^) associated with the smallest distance between filaments (1.0–1.2 mm), thereby generating greater impregnation homogeneity and reduced defect, breakage of fibres, and pore formation. In contrast, the printing space and thickness combination (Figure 7b) exhibits the highest flexural behaviour for a 0.41–0.45 mm printing thickness and 1.0–1.2 mm printing space. This behaviour occurs because a printing space larger than 1.2 mm induces an inter-filament porosity, decreasing the bond strength between the printed filaments.

Concerning the post-curing temperature and vacuum pressure combination (Figure 7c), the highest flexural strength comprises the range of −0.10 to −0.04 MPa for the vacuum and 160–185 °C for the cure temperature. The higher vacuum pressure increases the possibility of removing the gases and pores formed during the curing process, ensuring improvements in the reinforcement/matrix interface, which is directly proportional to the mechanical behaviours of the composite. Likewise, the proper temperature ensures the reduction of residual stresses during curing, as associated with the nature of the polymerization behaviour and crosslinking reactions of the matrix used.

The interaction between the processing parameters and flexural modulus is directly proportional to the flexural strength. The curves behave similarly to that described previously for the combinations of printing speed and space (Figure 7d), printing thickness and space (Figure 7e), and cure temperature and vacuum (Figure 7f). However, the variation between the results is associated with the nature of the modulus analysis, which presents a purely elastic behaviour, and the maximum stress analyzed that can present internal damage, for example, adhesive/cohesive microfractures and fibre ruptures.

The same analysis can be performed on the strain (Figure 8), in which the response surface exhibits a prediction behaviour for the intermediate parameter levels of the AM. All combinations can be found in the Appendix A. The strain behaviour is the opposite of those found for the flexural strength and modulus, considering that the deformation is greater with empty spaces between the printed filaments, thereby increasing the flexural capacity for lower printing thickness values and a higher printing speed and larger space (Figure 8a,b).
(12)$$s=0.27+5\times 10^{-4}P_{v}+1.45P_{s}-1.2\times 10^{-7}P_{v}^{2}-0.56P_{s}^{2}-2.4\times 10^{-4}P_{v}P_{s}$$
(13)$$s=0.4+0.6P_{s}-2.1t+3.3\times 10^{-12}P_{s}^{2}+3.1t^{2}-0.9P_{s}t$$
(14)$$s=-1.4+0.03T-6.1P-7.9\times 10^{-5}T^{2}-36.8P^{2}+0.02PT$$
where *s* (%) represents the flexural strain, *Pv* (in mm·min^−1^) printing speed, *Ps* (in mm) printing space, *t* (in mm) printing thickness, *T* (in °C) cure temperature, and *P* (in MPa) is the vacuum pressure.

The same behaviour is observed for the lower vacuum pressure, in which fewer pores are removed from the material, thereby maintaining a high deformation capacity (Figure 8c). However, the cure temperature level aiming for strain maximization has the same range as for flexural strength and modulus (160–185 °C), as the residual stress must be reduced to ensure the load transmission capacity during the flexural bending application.

The process optimization results are expressed in Table 4, which were based on RSM and ANN. The optimization criteria were to maximize the flexural strength and modulus and, consequently, minimise the deformation (considering the stiffness characteristic of the composite studied). The optimal solution was based on maximizing the desirability function. The difference between experimental results represents the error value obtained from the levels of the optimized parameters. The results of the desirability function were between 0.90 and 0.93%, considering the limitation of some parameter levels during processing. Nevertheless, the error of 0.13–2.57% indicates good reproducibility of the generated model. 

The ANN data optimization shows higher reliability with lower error than RSM. However, the predicted flexural strength, modulus, and strain errors were lower than 2.6%, ensuring engineering reliability. The diagrams of each parameter effect are exhibited in the Appendix A. Figure 9 exhibits the overlapping diagram of the optimization procedure based on input parameters with the addition of results from Table 4. This method maximizes flexural strength and modulus; meanwhile, the strain was minimised.

## 5. Conclusions

This study reflects the importance of statistical approaches to the processing parameters of CFRPs processed by AM. The ANN application demonstrated the possibility of mechanical prediction through manufacturing parameter level variations. After establishing the ANN parameters, different input data can be easily included in the trained neural network and the outputs can be appropriately obtained, resulting in a predictive method with high reliability and low error. For future works, the prediction procedure can be applied to different process parameters to predict flexural strength, modulus, and strain values not accessed experimentally through the proposed prediction models (ANN and RSM). 

The ANOVA methods treated the variance in the mechanical properties, evidencing a balanced contribution of the printing space, thickness, cure temperature, and vacuum as the main contributing factors. For future perspectives, the sequence of the main parameters to control defect formation and maximize mechanical properties are vacuum pressure > printing thickness > cure pressure = printing space > printing speed.

The prediction of the mechanical behaviour by the RSM confirmed the interaction among each processing parameter, allowing for the intermediate mechanical behaviours of each parameter level to be obtained. The mechanical behaviour tendency is a feasible method for determining the appropriate manufacturing parameter ratio(s) and optimizing the mechanical performance for specific applications, saving future project time and cost.

## Figures and Tables

**Figure 1 polymers-14-03668-f001:**
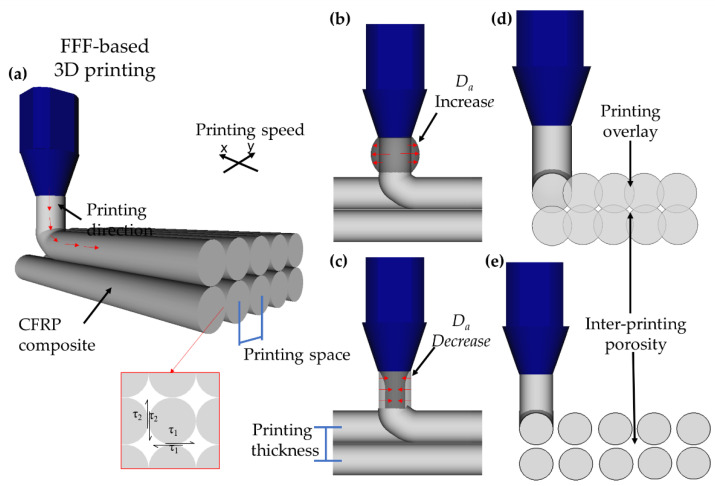
(**a**) Fused filament fabrication (FFF) scheme, (**b**) increase of printed filament diameter, (**c**) decrease of printed filament diameter, (**d**) printed filament overlay, and (**e**) inter-printing porosity (adapted from [8,20,21]).

**Figure 2 polymers-14-03668-f002:**
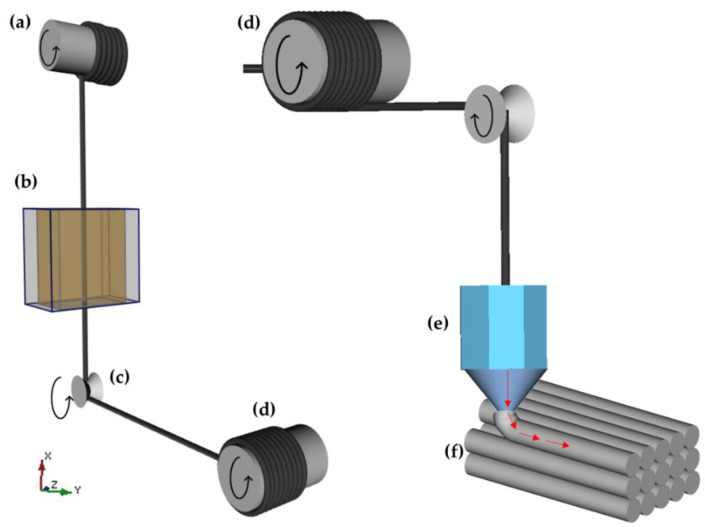
Scheme of FFF-based 3D printing: (**a**) dry fibre filament bobbin; (**b**) filament impregnation process; (**c**) steering pulleys; (**d**) wet fibre filament bobbin; (**e**) printing equipment; (**f**) printed continuous fibre-reinforced thermosetting polymer composites.

**Figure 3 polymers-14-03668-f003:**
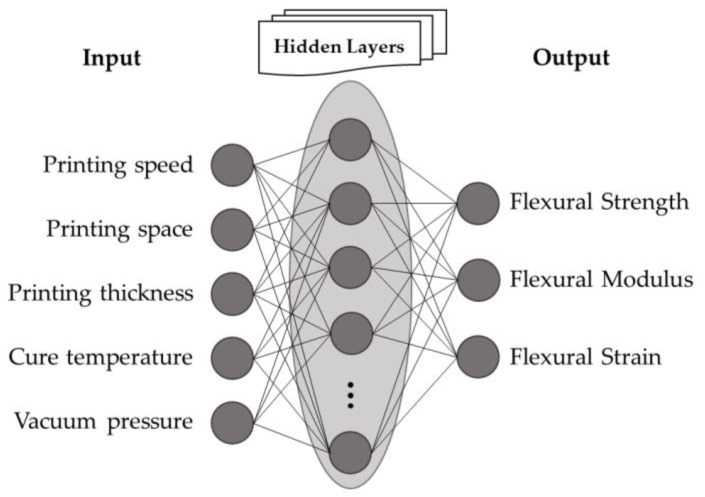
Artificial neural network (ANN) scheme correlation of AM processing parameters and mechanical behaviours.

**Figure 4 polymers-14-03668-f004:**
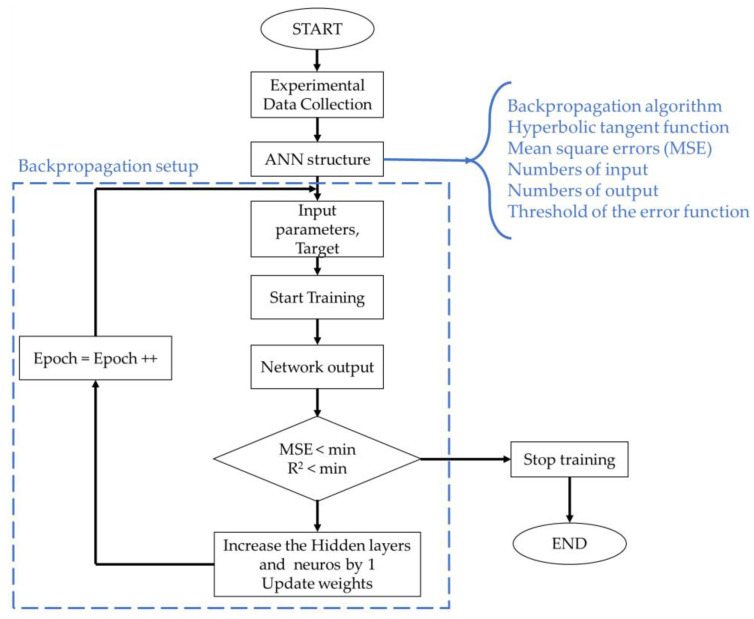
Schematic representation of neural networks training process.

**Figure 5 polymers-14-03668-f005:**
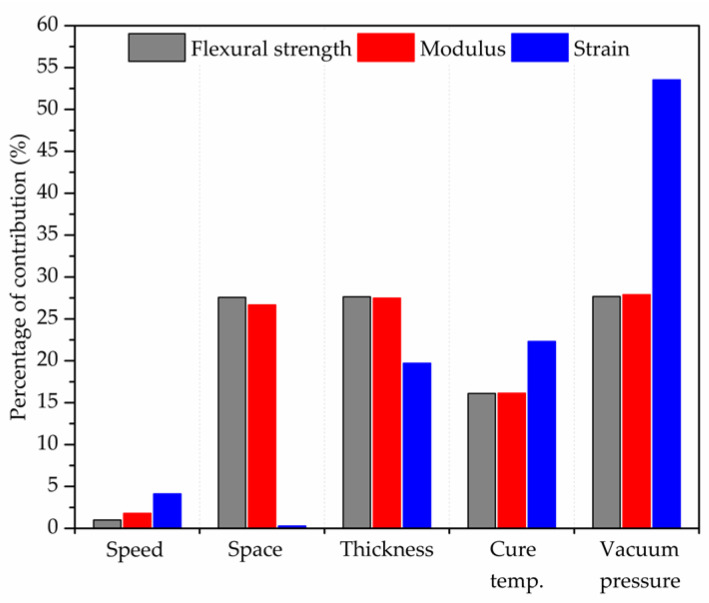
Percentage of contribution (PC) of processing parameters on mechanical behaviour.

**Figure 6 polymers-14-03668-f006:**
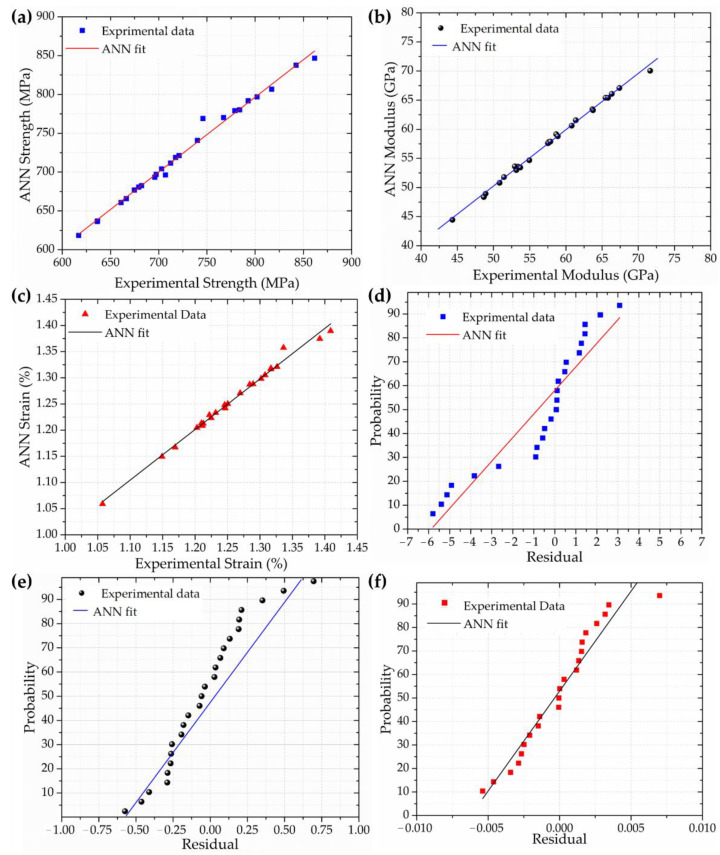
ANN regression plot: (**a**) flexural strength, (**b**) modulus, and (**c**) strain; ANN residual plot: (**d**) flexural strength, (**e**) modulus, and (**f**) strain.

**Figure 7 polymers-14-03668-f007:**
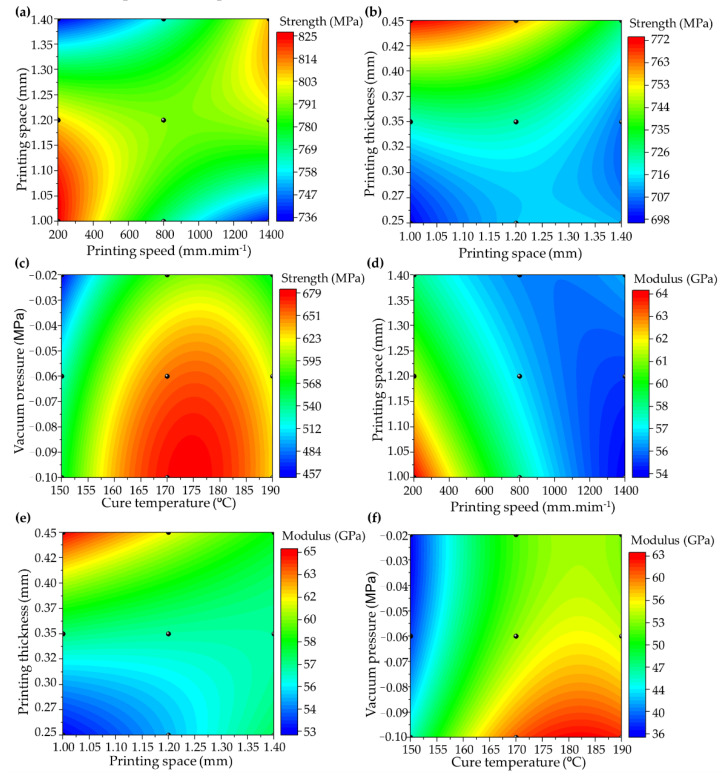
Response surface methodology (RSM) of flexural strength: (**a**) printing space vs. speed (Equation (6)), (**b**) printing thickness vs. space (Equation (7)), and (**c**) cure vacuum pressure vs. cure temperature (Equation (8)). RSM of flexural modulus: (**d**) printing space vs. speed (Equation (9)), (**e**) printing thickness vs. space (Equation (10)), and (**f**) cure vacuum pressure vs. cure temperature (Equation (11)).

**Figure 8 polymers-14-03668-f008:**
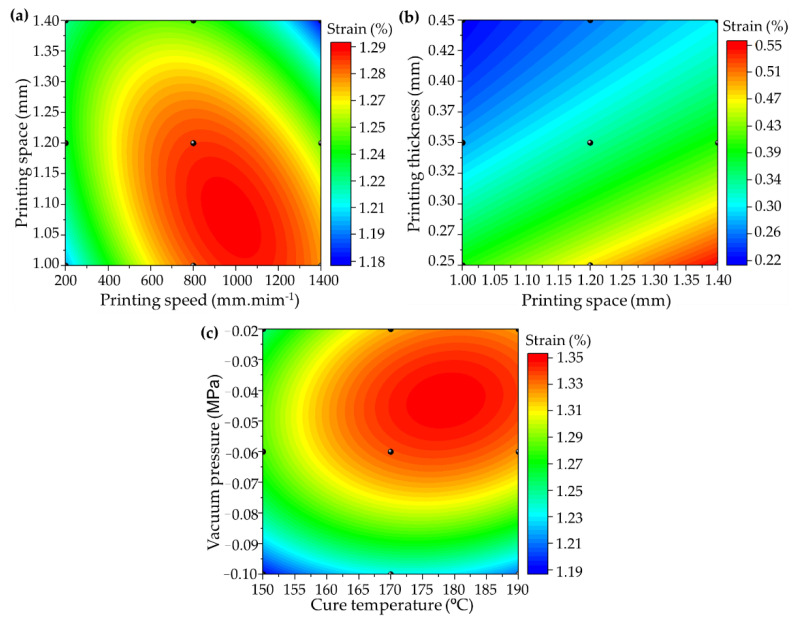
RSM of strain deformation: (**a**) printing space vs. speed (Equation (12)), (**b**) printing thickness vs. space (Equation (13)), and (**c**) cure vacuum pressure vs. cure temperature (Equation (14)).

**Figure 9 polymers-14-03668-f009:**
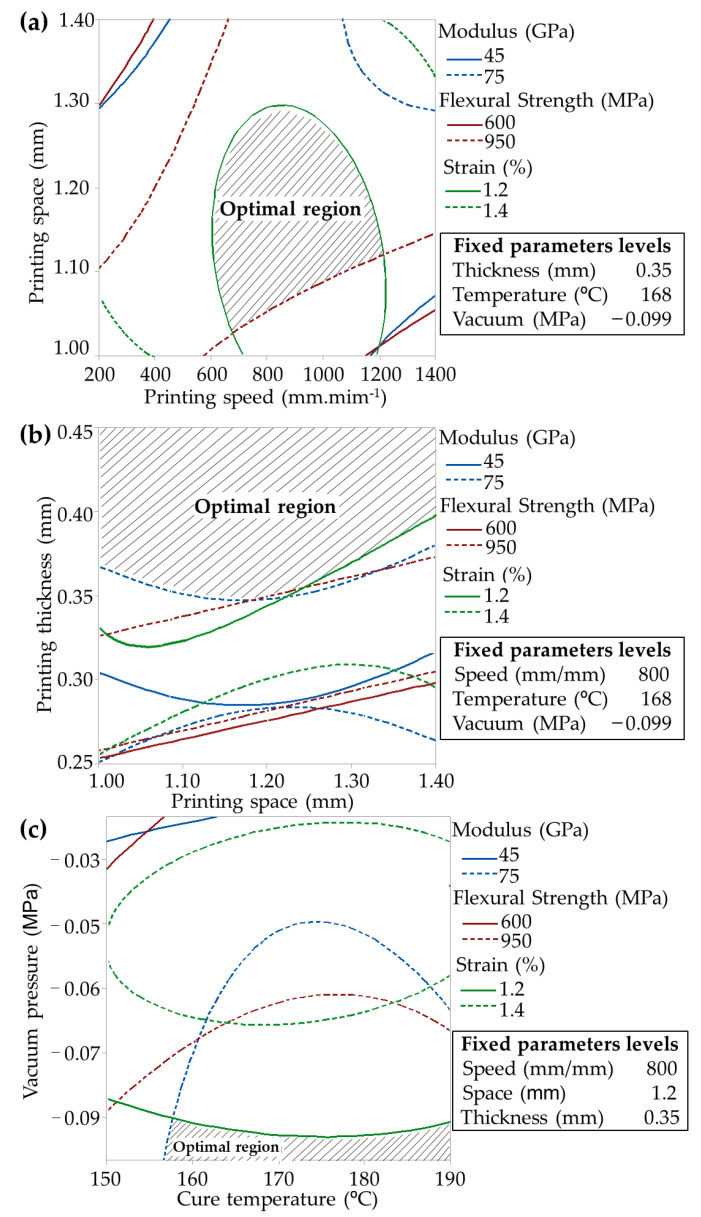
Overlapping plots of 3D RSM curves: (**a**) printing space against speed, (**b**) printing thickness against space, and (**c**) cure vacuum pressure against cure temperature.

**Table 1 polymers-14-03668-t001:** Experimental processing parameters levels L_25_ array (based on Refs. [17,42]).

Exp. No	Speed (mm·min^−1^)	Spacing (mm)	Thickness (mm)	Curing Temperature (°C)	Vacuum Pressure (MPa)
**1**	200	1.40	0.45	190	−0.10
**2**	500	1.20	0.30	180	−0.10
**3**	800	1.00	0.40	170	−0.10
**4**	1100	1.30	0.25	160	−0.10
**5**	1400	1.10	0.35	150	−0.10
**6**	200	1.30	0.40	180	−0.08
**7**	500	1.10	0.25	170	−0.08
**8**	800	1.40	0.35	160	−0.08
**9**	1100	1.20	0.45	150	−0.08
**10**	1400	1.00	0.30	190	−0.08
**11**	200	1.20	0.35	170	−0.06
**12**	500	1.00	0.45	160	−0.06
**13**	800	1.30	0.30	150	−0.06
**14**	1100	1.10	0.40	190	−0.06
**15**	1400	1.40	0.25	180	−0.06
**16**	200	1.10	0.30	160	−0.04
**17**	500	1.40	0.40	150	−0.04
**18**	800	1.20	0.25	190	−0.04
**19**	1100	1.00	0.35	180	−0.04
**20**	1400	1.30	0.45	170	−0.04
**21**	200	1.00	0.25	150	−0.02
**22**	500	1.30	0.35	190	−0.02
**23**	800	1.10	0.45	180	−0.02
**24**	1100	1.40	0.30	170	−0.02
**25**	1400	1.20	0.40	160	−0.02

**Table 2 polymers-14-03668-t002:** ANOVA results for flexural strength, modulus, and strain.

*Flexural Strength*	*F*	*p*-Value	*F_critical_*
Speed	15.95	2.10 × 10^−2^	3.95
Space	43.99	3.31 × 10^−78^	3.95
Thickness	44.09	2.99 × 10^−78^	3.95
Cure temperature	25.67	5.90 × 10^−68^	3.95
Vacuum pressure	44.15	2.84 × 10^−78^	3.95
** *Flexural Modulus* **	** *F* **	***p*-Value**	** *F_critical_* **
Speed	21.13	2.39 × 10^−25^	3.95
Space	312.13	1.17 × 10^−71^	3.95
Thickness	321.74	2.99 × 10^−78^	3.95
Cure temperature	188.92	3.40 × 10^−62^	3.95
Vacuum pressure	326.56	1.62 × 10^−72^	3.95
** *Flexural Strain* **	** *F* **	***p*-Value**	** *F_critical_* **
Speed	24.45	2.14 × 10^−27^	3.95
Space	5.75	6.57 × 10^−05^	3.95
Thickness	416.94	3.53 × 10^−77^	3.95
Cure temperature	532.34	7.27 × 10^−82^	3.95
Vacuum pressure	117.45	3.73 × 10^−97^	3.95

**Table 3 polymers-14-03668-t003:** Average results of experimental and ANN prediction of flexural mechanical behaviour.

No	Experimental Data			ANN Prediction Data			Average Error (%)
	Flexural Strength (MPa)	Flexural Modulus (GPa)	Strain (%)	Flexural Strength (MPa)	Flexural Modulus (GPa)	Strain (%)	
**1**	666.36	54.95	1.21	665.78	54.68	1.21	0.19
**2**	712.34	58.88	1.21	711.49	58.81	1.21	0.18
**3**	697.50	57.58	1.21	697.02	57.65	1.21	0.14
**4**	682.40	51.45	1.33	682.51	51.80	1.32	0.37
**5**	702.97	53.38	1.32	704.14	53.59	1.32	0.22
**6**	767.16	58.66	1.31	770.24	59.15	1.31	0.48
**7**	861.60	71.63	1.20	846.60	70.06	1.20	1.39
**8**	792.85	63.66	1.25	791.94	63.46	1.25	0.21
**9**	817.12	66.34	1.23	806.71	66.08	1.23	0.60
**10**	721.16	57.85	1.25	721.23	57.89	1.24	0.15
**11**	801.94	65.47	1.22	796.80	65.41	1.22	0.29
**12**	842.47	67.38	1.25	837.55	67.10	1.25	0.34
**13**	636.22	48.66	1.31	636.32	48.37	1.30	0.27
**14**	783.97	60.80	1.29	780.15	60.62	1.29	0.30
**15**	778.96	63.74	1.22	779.13	63.28	1.23	0.44
**16**	717.46	61.36	1.17	718.89	61.55	1.17	0.23
**17**	740.25	57.64	1.28	740.73	57.83	1.29	0.21
**18**	616.99	44.32	1.39	618.43	44.45	1.37	0.60
**19**	679.31	50.83	1.34	680.58	50.80	1.36	0.61
**20**	745.86	52.94	1.41	769.05	53.63	1.39	1.90
**21**	660.87	57.51	1.15	660.69	57.60	1.15	0.07
**22**	696.00	65.82	1.06	693.32	65.41	1.06	0.39
**23**	674.68	53.15	1.27	676.85	53.00	1.27	0.23
**24**	636.58	48.90	1.30	637.13	48.93	1.30	0.14
**25**	707.05	53.70	1.32	696.22	53.44	1.32	0.68

**Table 4 polymers-14-03668-t004:** Process parameter optimization.

Parameters	Optimal Inputs		
	RSM	ANN	
**Speed** (mm·min^−1^)	800	796	
**Space** (mm)	1.20	1.20	
**Thickness** (mm)	0.35	0.35	
**Temperature** (°C)	168	166	
**Vacuum** (MPa)	−0.099	−0.096	
**Desirability**	0.90	0.93	
**RSM**	*Flexural Strength* (MPa)	*Modulus* (GPa)	*Strain* (%)
**Predicted values**	958.81	72.87	1.29
**Error**	0.62%	−1.28%	−2.57%
**ANN**	*Flexural Strength* (MPa)	*Modulus* (GPa)	*Strain* (%)
**Predicted values**	954.15	71.52	1.34
**Error**	0.13%	−0.60%	1.26%

## Data Availability

The data will be made available upon request to the corresponding author.

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
