# Peer review of "Prediction of Bending Properties for 3D-Printed Carbon Fibre/Epoxy Composites with Several Processing Parameters Using ANN and Statistical Methods"

_polymers, 2022, doi:10.3390/polym14173668_

Round 1

Reviewer 1 Report

Dear Authors

The paper was fairly prepared, and I believe that the work carried out can be of scientific interest to the many readers of the journal. The following though, comments to improve the readability of the paper;

-          The abstract could be improved. The statement "The predicted results present high reliability and low error level compared to the experimental results" may need to be revisited. The experimental results should be true data; thus, the predicted data could not be truer than the actual findings. Need to revise.

-          Methods –to explain further how the ANN model is developed and how many datasets were used to develop the modelled.

-          Pls also show the example of the dataset used for the training of the ANN model.

-          Please also include how the data processing and classification were performed when the model was developed.

-          What are the test protocol and the model's classification, and is there a limitation to the model?

-          Also, the authors to include a flowchart of the ANN model to predict the flexural properties.

-          How is the model validated?

-          Section 3.1 should be removed or moved to earlier sections; - intro. It's not results and discussion.

-          May want to relook into the conclusion after the amendment suggested.

Author Response

Reviewer #1:

The paper was fairly prepared, and I believe that the work carried out can be of scientific interest to the many readers of the journal. The following though, comments to improve the readability of the paper:

- The abstract could be improved. The statement "The predicted results present high reliability and low error level compared to the experimental results" may need to be revisited. The experimental results should be true data; thus, the predicted data could not be truer than the actual findings. Need to revise.

We revised and corrected this sentence and checked the abstract to avoid this kind of misunderstanding.

- Methods –to explain further how the ANN model is developed and how many datasets were used to develop the modelled.

We have extended the ANN methodology explanation to add this information. Also, additional information was included in the Supplementary Material.

- Pls also show the example of the dataset used for the training of the ANN model.

The performance of training ANN is shown in Figure S1 as supplementary material.

- Please also include how the data processing and classification were performed when the model was developed.

We have extended the ANN methodology explanation to add this information.

- What are the test protocol and the model's classification, and is there a limitation to the model?

We have extended the ANN methodology explanation to add this information. The main limitations are the dependency on the input data and sensitivity to noisy data. However, the presented material showed good reproducibility with low standard deviation, in which all main factors were considered as input data.

- Also, the authors to include a flowchart of the ANN model to predict the flexural properties.

We added the flowchart, as suggested.

- How is the model validated?

The model was validated based on mean square errors (MSE) of 0.0001 and R2 > 0.99 to ensure better interaction with experimental data.

- Section 3.1 should be removed or moved to earlier sections; - intro. It's not results and discussion.

We agree, and we change the previous section's location.

We thank you for your excellent comments and recommendations. All suggestions have been considered in the revised version of the manuscript.

Reviewer 2 Report

The authors reported an interesting work predicting the process-structure-behavior properties of CFRPs process during 3d printing. The work is well done and of general interesting. However, there are some reference missing, and some results/conclusion need further clarification. Therefore, a major revision is needed, and the following question must be addressed.

1. Figure 4, the cure temperature seems to be a big factor for all three properties? How is the cure temperature influences the strength, modulus, and strain? Some discussion could be added.

2. Figure 4, why does space have little impact on strain? Considering the modulus and strain are related. Some discussion could be added.

3. Is there any hypothesis that the error in Table 3, entry 20 is the way larger than the others?

4. Line 45, the introduction about AM process should be further expanded. The advantages of 3D printing should be discussed, and recent other methods to fabricate carbon fiber(cnt)/polymer composites (https://doi.org/10.1039/D1PY00705J; https://doi.org/10.1016/j.apmt.2017.04.003) should be cited and briefly discussed.

5. Some sentences are missing references.

a. “The printing overlay can strengthen the bonding to enhance the shear between the printed filaments, ensuring high mechanical  performance”

b. “The curing temperature affects the viscosity, polymerization rate, and crosslinking reactions (i.e., kinetic energy), similar to the case with conventional thermoset composite manufacturing”

6. Maybe I missed, but I did not see any explanation of the letters in equation 5-12 nor the discussion about those equations.

Author Response

Reviewer #2:

The authors reported an interesting work predicting the process-structure-behavior properties of CFRPs process during 3d printing. The work is well done and of general interesting. However, there are some reference missing, and some results/conclusion need further clarification. Therefore, a major revision is needed, and the following question must be addressed.

  1. Figure 4, the cure temperature seems to be a big factor for all three properties? How is the cure temperature influences the strength, modulus, and strain? Some discussion could be added.

The curing temperature affects the curing speed of the thermoset matrix, which can generate residual stress and incomplete bonds (voids) at an inappropriate temperature. Thus, the stresses generated during the curing stage will directly influence the mechanical behavior of the material, mainly at intra-imprint interfacial strength. We added this explanation to the manuscript.

  1. Figure 4, why does space have little impact on strain? Considering the modulus and strain are related. Some discussion could be added.

The printing spacing on the vertical axis (printing thickness) can result in a spacing between the layers along the thickness, generating a state of compressive deformation during the 3-point bending test, increasing the contribution of the printing thickness parameter. On the other hand, printing space is related to the space between the filaments on the horizontal plane axis, which has less impact on the flexural force distribution and could demonstrate different contributions to other loading applications.

  1. Is there any hypothesis that the error in Table 3, entry 20 is the way larger than the others?

All errors were well below the standard deviation between testing repetitions, indicating that the presented and questioned variations are related to the intrinsic standard deviation of the proposed experiments than any specific combination that could provide greater or lower error during modelling.

  1. Line 45, the introduction about AM process should be further expanded. The advantages of 3D printing should be discussed, and recent other methods to fabricate carbon fiber(cnt)/polymer composites (https://doi.org/10.1039/D1PY00705J; https://doi.org/10.1016/j.apmt.2017.04.003) should be cited and briefly discussed.

We have revised the introduction and included the suggested references.

  1. Some sentences are missing references.

      a. “The printing overlay can strengthen the bonding to enhance the shear between the printed filaments, ensuring high mechanical performance”

      b. “The curing temperature affects the viscosity, polymerization rate, and crosslinking reactions (i.e., kinetic energy), similar to the case with conventional thermoset composite manufacturing”

We carefully revised the absence of references in the above sentence and added the suggested Refs.

  1. Maybe I missed, but I did not see any explanation of the letters in equation 5-12 nor the discussion about those equations.

They were detailed in material and methods (section 3.4). However, we added it in the results sections closer to those equations.

Thank you for your excellent comments and recommendations. All suggestions have been considered in the revised version of the manuscript.

Reviewer 3 Report

In the current study, Monticeli et al., have employed two models (if I understand correctly, they are inter-connected, the ANN model would train parameters based on experimental results and transfers these fitted parameters to the RSM model) for predicting the effect of multiple factors, including vacuum pressure, printing speed, curing temperature, printing space, and thickness, on the mechanical properties of carbon fiber-reinforced polymers. The paper is well-written, easy to read, and would bring broad interests to the relevant readers. Before the paper can be accepted for publishing on Polymers, could the authors address the following issues?

1.     Through what procedure that the beta values in Eq.3 are obtained, and therefore used in Eq. 4-12? The authors simply mentioned that the predicted data from the ANN were used by Eq.3, however, which category of parameters in Eq.3 are from the ANN prediction remains unclear. Thus, more simulation details have to be provided. 

2.     Why do the authors still consider that the printing speed as an effective variable as the contribution of printing speed is negligible comparing to other parameters (Figure 4)?

3.     The R2 for Figure 5d-f are not mentioned, and could the authors explain more about why the fittings for strain are better?

4.     The black points on Figure 6 and 7 are not explained.

5.     Figure 7 has a line/border on the left, please check the figure format.

6.     I didn’t see any software/code in ESI rather than some snapshots from OriginLab (as indicated by Line 375), so where is the software the authors were talking about? 

Author Response

Reviewer #3:

In the current study, Monticeli et al., have employed two models (if I understand correctly, they are inter-connected, the ANN model would train parameters based on experimental results and transfers these fitted parameters to the RSM model) for predicting the effect of multiple factors, including vacuum pressure, printing speed, curing temperature, printing space, and thickness, on the mechanical properties of carbon fiber-reinforced polymers. The paper is well-written, easy to read, and would bring broad interests to the relevant readers. Before the paper can be accepted for publishing on Polymers, could the authors address the following issues:

  1. Through what procedure that the beta values in Eq.3 are obtained, and therefore used in Eq. 4-12? The authors simply mentioned that the predicted data from the ANN were used by Eq.3, however, which category of parameters in Eq.3 are from the ANN prediction remains unclear. Thus, more simulation details have to be provided. 

The procedure to generate Eq. 3 constants was carried out with the experimental data and ANN data (assuming the low error was found between ANN and experimental data, within the standard deviation). The objective was to increase the number of repetitions, ensuring the refinement of the statistical regression. The text has been updated.

  1. Why do the authors still consider that the printing speed as an effective variable as the contribution of printing speed is negligible comparing to other parameters (Figure 4)?

This occurs because the printing speed variation can generate the stretching or contraction of the printing section (changing printing diameter), as explained in Section 2. However, this variation can be supplied by controlling the horizontal and vertical spacing between the filaments, reducing the experimental influence (or percentage of contribution) of the printing speed compared with the proposed parameters.

  1. The R2 for Figure 5d-f are not mentioned, and could the authors explain more about why the fittings for strain are better?

The coefficient of determination presented a variation between R2 = 0.91 – 0.97, keeping the results with R2 > 0.91.
The fittings for strain are not better than those presented for flexural strength and modulus. Strain represents an opposite behaviour for RSM than those found in strength and modulus, in which higher values of modulus and strength occur when material presents lower strain (considering the stiffness and brittle behaviour of composite material). In addition, the error shown in Table 4 presents a higher error for strain than those found for flexural strength and modulus. However, the error found is within the standard deviation of the experimental values, which makes it insignificant.

  1. The black points on Figures 6 and 7 are not explained.

We added this explanation in the text.

  1. Figure 7 has a line/border on the left, please check the figure format.

We fixed that.

  1. I didn’t see any software/code in ESI rather than some snapshots from OriginLab (as indicated by Line 375), so where is the software the authors were talking about? 

We have added more information about the software used and the appropriate parameters for ANN analysis.
We have edited OriginLab with ANN function to predict proposed mechanical behaviour following steps sent in the supplementary material. In addition, we also are adding the OriginLab software for this use.

Thank you for your excellent comments and recommendations. All suggestions have been considered in the revised version of the manuscript.

Round 2

Reviewer 1 Report

The authors have addressed all the comments fairly, and the manuscript can be accepted for publication.

Author Response

Thank you very much for your assessment of our manuscript.

Reviewer 2 Report

The authors took efforts to improve the manuscript, but still not enough. There are still many questions. Moreover, they claimed in their response that they made the revision for some questions, but they actually did not make the revision into the manuscript! Please do solve the following questions before I can consider it for publication.

1.  For my question 2, the authors'  did not directly answer my question about "why does strain have little impact on strain but have big impact on modulus".. The authors answer is about space influence on "vertical axis" and "horizontal plane axis", but not to my question. Please give a better explanation. 

2. For my question 1, the authors said " the thermoset matrix"   can generate residual stress and incomplete bonds (voids) at an inappropriate temperature. Any support for the sentence? Give the related reference is a must.

3. Similarly, for my question 2, the authors answered "printing space is related to the space between the filaments on the horizontal plane axis, which has less impact on the flexural force distribution and could demonstrate different contributions to other loading applications." The first sentence is easy to understand, but why it has less impact on flexural force? Any support? Any reference? Besides, "could demonstrate different contributions to other loading applications" is not specific and pointless. What applications ?  How different? Please either be more specific or delete this sentence.

4. For my question 4, the authors did some revision, and answered " The 3D printing with abrasives, such as carbon nanotubes and graphene". However, the reference is still lacking, although they said they add them in the response. Please add the the references/examples for "carbon nanotubes" and "graphene", including the suggested references last time.

5. The authors claimed in response they added reference for  " The curing temperature affects the viscosity, polymerization rate, and crosslinking reactions (i.e., kinetic energy), similar to the case with conventional thermoset composite manufacturing”, but they did not! Please check the whole manuscript again for the references and do not make this type of carless mistakes.

Author Response

Thank you very much for your assessment of our manuscript.

The authors took efforts to improve the manuscript, but still not enough. There are still many questions. Moreover, they claimed in their response that they made the revision for some questions, but they actually did not make the revision into the manuscript! Please do solve the following questions before I can consider it for publication.

We have corrected and highlighted all responses made to the reviewer's questions in the manuscript. We respectfully hope that the final version is suitable for publication.

  1. For my question 2, the authors' did not directly answer my question about "why does strain have little impact on strain but have big impact on modulus". The authors answer is about space influence on "vertical axis" and "horizontal plane axis", but not to my question. Please give a better explanation.

We focused on the first answer regarding the printing space and thickness parameter because they are the parameter that influences the response. What happens in this analysis is the measurement of each parameter contribution generated in the response. This occurs through the response standard deviation (strength, modulus, and strain) as a function of the parameter levels variability. This means that all parameters influence the result, but comparatively, they can present different contributions that result from the variance of the results of mechanical properties.

To answer the question, let us deal with the basic concept of the stress-strain graph. With the variation of the processing parameters, we have differences in the material's mechanical properties, as shown in the work. However, the spacing generated between printed filaments results in a vertical reduction of the maximum load capacity (strength), modifying the angle stress-strain curve (i.e., modulus), and maintaining the strain levels close to each other since the material has a brittle characteristic. Thus, the percentage of contribution for strain is lower than the strength and modulus variance. On the other hand, strain is affected by curing parameters since they guarantee greater binding efficiency of macromolecules.

  1. For my question 1, the authors said " the thermoset matrix" can generate residual stress and incomplete bonds (voids) at an inappropriate temperature. Any support for the sentence? Give the related reference is a must.

We had added the appropriate Refs. [62-64].

  1. Similarly, for my question 2, the authors answered "printing space is related to the space between the filaments on the horizontal plane axis, which has less impact on the flexural force distribution and could demonstrate different contributions to other loading applications." The first sentence is easy to understand, but why it has less impact on flexural force? Any support? Any reference?
    Besides, "could demonstrate different contributions to other loading applications" is not specific and pointless. What applications?  How different? Please either be more specific or delete this sentence.

The printing space and thickness contributions to response are analyzed as a comparative analysis, in which contribution is based on normalized behaviour of all levels. We observed this behaviour based on experimental results since thickness space leads to the possibility of a compression state of compressive deformation – behaviour not observed in printing space (horizontal). However, we found similar results as shown in Ref. [51].
We have removed the second sentence since it does not add important information about the work.

  1. For my question 4, the authors did some revision, and answered " The 3D printing with abrasives, such as carbon nanotubes and graphene". However, the reference is still lacking, although they said they add them in the response. Please add the the references/examples for "carbon nanotubes" and "graphene", including the suggested references last time.

We respectfully appreciate the article suggestions. Both suggested references were added. As a matter of fact, we have included some references and enhanced the introduction with this focus.

  1. The authors claimed in response they added reference for "The curing temperature affects the viscosity, polymerization rate, and crosslinking reactions (i.e., kinetic energy), similar to the case with conventional thermoset composite manufacturing”, but they did not! Please check the whole manuscript again for the references and do not make this type of carless mistakes.

In fact, the references had been added only at the end of the paragraph. We understand that the distance between the citation and the text can cause this misunderstanding, so we corrected the reference position.

The authors reported an interesting work predicting the process-structure-behavior properties of CFRPs process during 3d printing. The work is well done and of general interesting. However, there are some reference missing, and some results/conclusion need further clarification. Therefore, a major revision is needed, and the following question must be addressed.

We have added all suggestions and recommendations to the manuscript. 

The authors deeply appreciate your re-assessment of our manuscript.

Round 3

Reviewer 2 Report

good to publish now